# Demonstrating Task-Aware Saliency for 3D Shapes

Category: Research

## ABSTRACT

The concept of task-aware saliency has been demonstrated for 2D images. The idea of this concept is that for different tasks, where people look or focus their attention (i.e. saliency) on 2D images is different. However, this concept has not been demonstrated for 3D shapes. In this paper, we collect data on where people look on 3D shapes while performing different tasks (e.g. classifying a shape, deciding how aesthetic a shape is, or describing a shape). We then display the data as task-aware saliency maps on the 3D shapes, and demonstrate that where people look on 3D shapes as they are performing different tasks is different.

**Index Terms:** Computing methodologies—Computer graphics—Graphics systems and interfaces—Perception; Computing methodologies—Computer graphics—Shape modeling—Mesh models

## 1 INTRODUCTION

The concept of considering where people look and pay attention to has been important in many areas, for example in entertainment (e.g. to make more appealing games and films), advertising (e.g. to create ads that maximize profit), and health (e.g. to better understand patients with dementia). We consider the concept of where people look from the computing perspective, where it is commonly described as visual saliency and has been explored for images [1, 2, 16, 17] and 3D meshes [19, 22, 25]. Moreover, it has been known since the work of Yarbus [51] that the viewer task affects where people look on images. In other words, where people look or focus their attention to (i.e. saliency) as they perform different tasks on 2D images is different (across the tasks). This concept is referred to as "task-aware saliency", and it has been demonstrated on 2D images [29], webpages [52], and interactions in virtual environments [37].

However, for 3D shapes, visual saliency that takes into account the viewer task has not been considered before. This is an important problem as intuitively when one looks at a virtual 3D shape, there is often some reason, purpose, or task. Our hypothesis is that, for 3D shapes, the viewer task will affect where people look. For example, for the task of shape classification, one may look quickly perhaps near the center of a shape to decide what real-world object it is. For the task of judging the aesthetics of a 3D shape, one may spend more time observing the edges or boundaries since the curved parts of a shape tend to make it more aesthetic [12, 28]. Or for the task of describing a 3D shape in words, one may need time to observe each part of the shape in detail.

In this paper, our goal is to demonstrate that this concept of "task-aware saliency" is also true for 3D shapes. We first collect data on where people look on 3D shapes for different tasks. We get 3D shapes from the ShapeNet dataset [8], and we have three shapes from each of four classes: club chairs, tables, lamps, and an "abstract" class. For each 3D shape, we display it with a multiple-view (3 views) image representation. To collect our data, we ask our participants to observe each shape while performing each of four tasks: classifying the shape (state the shape class or name), deciding how aesthetic the shape is (give a score between 1 and 5), describing the shape in words (by stating words), and deciding the materials that the shape can be made of if it were a real-world object (by stating words). To collect visual saliency data, we implement a one-person clicking interface [18] where the clicked locations on blurred images are salient. So we initially blur the multi-view images

representing each 3D shape and show them to participants. We then ask the participants to click on them to unblur circular regions (these correspond to salient regions). The participants are asked to click as much as they need in order to perform the given task well.

After the data collection process, we combine the data for many participants and display them as "task-aware saliency maps" directly on the 3 multiple-view image representation of each 3D shape where the data was originally collected. We then compare these saliency maps for each shape *across the tasks*. The main result of this paper is that we demonstrate that where people look on 3D shapes as they perform different tasks is different.

## 2 RELATED WORK

### 2.1 Image Saliency

There exist much previous works in the topic of image saliency [14, 16] and a complete review is beyond the scope of this work. We refer the reader to some surveys for more detail [1, 2, 17]. Among previous works, we are inspired by methods to collect saliency data [4, 18] and to perform evaluation of saliency results [7]. In general, our consideration of 3D shapes (and mesh saliency) is different from images (and image saliency) in that "the nature of the problem is different" [25]. Images are regular 2D grids of colors and often contain environments or objects. In contrast, we focus on each 3D shape at a time and only consider its geometry with no colors and textures. Each 3D shape is an irregular set of 2D polygons in 3D space and usually represents a single well-defined object (e.g. chair, table, lamp).

### 2.2 Task-based Image Saliency

There are previous works for image saliency where the viewer is given some task, and the relations between the task and where people look are studied. These are for various tasks including: seven tasks while viewing a painting in Yarbus' seminal work (e.g. estimate material circumstances of family, give ages of people) [3, 51]; search tasks on images [53] (e.g. search for car in image [30]); counting the number of people in an image [33]; free viewing, saliency search task, or cued object search task [20]; trace lines of geometrical figures (e.g. rectangles, triangles, lines) or count number of straight lines [51]; "perception of pictures during reading, during perception of optical illusions, and during comparison of distances" [51]; three tasks for visualizing charts where an example task is: retrieve value of a specific data element [36]; and also the task of playing video games (i.e. for sequences of images) [5, 34]. Despite these task-based image saliency works, there is no previous work exploring task-based saliency for 3D shapes.

### 2.3 Mesh Saliency

In addition to the concept of image saliency, the idea of saliency was extended to 3D shapes or meshes [22]. Some examples include: detecting perceptually salient ridges and ravines on polygon meshes [15, 47], identifying distinctive regions of a mesh [39], comparing the computed mesh saliency with captured human eye movements [19], considering local [23, 31] and/or global features [41, 49], and tracking human fixations on physical 3D objects (i.e. 3D printed from virtual models) [45, 46]. There are also other examples that identify salient features in different ways [11, 40, 43]. Furthermore, there are concepts of saliency that are different from just visual saliency: Schelling salient points are selected with a coordination game [10], human-centered saliency is based on how a human uses

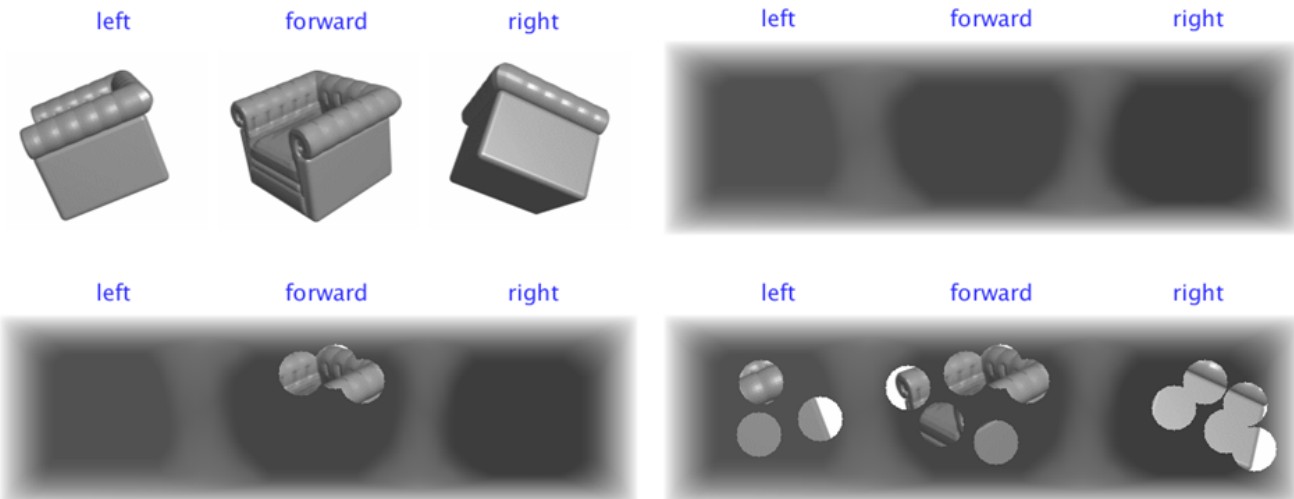

Figure 1: Top Left: The 3 views for an example 3D shape that is of the club chair class. These are not shown to the participants. Top Right: The Gaussian blurred multiple-view images that are initially shown to participants. Our clicking interface [18] allows participants to click on regions that they wish to observe. Bottom Row: The images with unblurred regions, where the center of each circular region is the user-clicked location.

an object [26], and tactile mesh saliency is based on how a human grasps, presses, or touches an object [21].

### 2.4 Perception of 3D Shapes

Existing works include the perception of 3D shape viewpoints [38], visual quality [6], and semantic attributes [9]. There is also previous works in perceiving 3D shapes themselves from depth cues [48] and visual and tactile observations [44]. Much work exists in the area of 3D shape perception [35], and we focus on one novel and specific problem in this paper.

### 2.5 Crowdsourcing of Human Perceptual Data

Our data collection framework is inspired by works that collect crowdsourced data (e.g. 2D clip art style [13], fonts [32], and 3D shapes [21, 24, 27]). There is also work in crowdsourcing 3D shape saliency data with a 2-player game method [50]. While crowdsourcing is not the focus of our work, it allows for the easy and useful collection of saliency data [18, 50].

## 3 COLLECTING TASK-AWARE SALIENCY DATA FOR 3D SHAPES

The data collection step collects data on where people look on the surfaces of 3D shapes as they perform different tasks. We refer to this as the "task-aware saliency" data.

We collect shapes from the ShapeNet dataset [8]. We have 12 shapes: 3 shapes from each of 4 classes. The classes are: club chairs, tables, lamps, and an "abstract" class. For each 3D shape, we display it as a multiple-view (3 views) image representation. This kind of multiple-view representation has been successfully used in previous works [21, 42]. We have a forward-looking (that is slightly-slanted) view, a view from approximately the left side of the shape, and a view from approximately the right side. The viewpoints are manually chosen. Figure 1 top left shows an example of the 3 views for a 3D shape that is of the club chair class.

The key idea for collecting the "task-aware saliency" data is to ask participants to observe a 3D shape while performing a task. We have 4 tasks: classifying the shape (state the shape class or name), deciding how aesthetic the shape is (give a score between 1 and 5), describing the shape in words (by stating words), and deciding the materials that the shape can be made of if it were a real-world object (by stating words). Participants were asked to observe each

shape as much as they need to perform each given task well. We have participants naturally perform each task without them knowing that the saliency aspect is what we are exploring (instead of just the responses from the tasks themselves).

In order to track where participants observed, we implement a one-person clicking interface [18] where the clicked locations on blurred images are salient. So we initially perform Gaussian blurring of the multiple-view images representing each 3D shape and show them to participants. We then ask the participants to click on locations that they wish to observe. Clicking on the images will unblur circular regions (these correspond to salient regions). Figure 1 top right shows an example of the Gaussian blurred multiple-view images. Figure 1 bottom row shows examples of the images with unblurred regions, where the center of each circular region is the user-clicked location.

For each 3D shape and task, we collect data for 15 participants. The participants were found at our university. Each participant was given instructions and a consent form at the start. Their participation was voluntary, and the participants were told they could stop if and whenever they wish. Each participant took an average of 17 seconds per 3D shape and task.

## 4 RESULTS

After the data collection process, we combine the data for all the participants for each 3D shape and task and display them as "task-aware saliency maps" directly on the 3 multiple-view image representation where the data was originally collected. We then compare these saliency maps for each shape *across the tasks*.

### 4.1 Displaying Task-Aware Saliency Maps

From the data collection step, we have a "task-aware saliency" map for each participant and each 3D shape and task. We represent this internally as a 2D grid of integer values, where higher values are more salient. From each user-clicked location, we have a circular region of values added to this map. The center of this region has a large value, followed by gradually decreasing values as we move further away from the center. We combine these "task-aware saliency" maps from many participants. For each 3D shape and task, we do so by adding and then normalizing the 2D grid of values for the participants. We then convert these values to the Matlab Jet colormap,

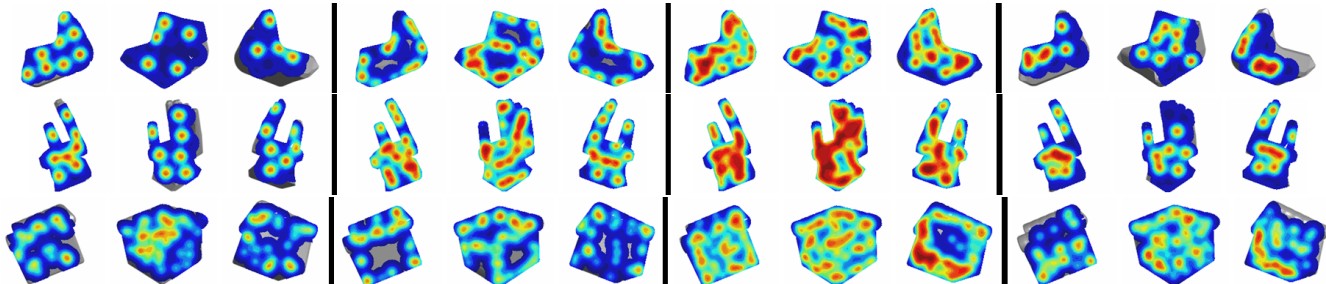

Figure 2: Results for 3 club chairs: Each row shows the results for a club chair. The second shape is a special case of a hand-shaped chair. Each column (separated by the vertical black lines) shows the 3 views of the 3D shape. The 4 columns show the saliency maps for the 4 tasks: classifying the shape (state the shape class or name), deciding how aesthetic the shape is (give a score between 1 and 5), describing the shape in words (by stating words), and deciding the materials that the shape can be made of if it were a real-world object (by stating words).

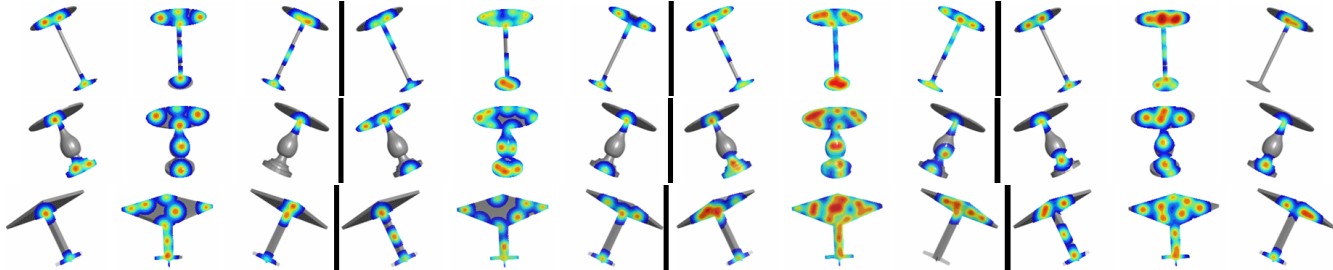

Figure 3: Results for 3 tables: Each row shows the results for a table. Each column (separated by the vertical black lines) shows the 3 views of the 3D shape. The 4 columns show the saliency maps for the 4 tasks (see Figure 2 or the text for more details).

which are blue-to-red colors (for low-to-high values). We display these colors (with some transparency) directly on top of the original 3 multi-view images. This is our combined "task-aware saliency" maps for each 3D shape and task. The figures (of our results) below visualize these combined "task-aware saliency" maps.

### 4.2 Qualitative Comparisons of Task-Aware Saliency Maps

The goal here is to get an intuition of what participants looked at, specifically *across the different tasks*, to gain an understanding of the differences (if any) across the tasks. We perform comparisons of our "task-aware saliency" maps, and observe how they are different and any possible reasons for why they are different.

In Figure 2, for the first row or first club chair, the first saliency map (for classifying the shape), has a relatively small number of red or salient parts. The second saliency map (for deciding shape aesthetic) has salient parts mostly around the edges. The third saliency map (for describing shape) has salient parts almost everywhere. The fourth saliency map (for deciding the materials the shape can be made of) has salient parts near the flat parts of the chair. For the second shape, it is a special case of a hand-shaped chair. The first saliency map has a relatively small number of red or salient parts (although more than the first chair as the second chair is a more complex shape). The second saliency map has salient parts near the edges (although this shape has many edges). The third saliency map has salient parts almost everywhere. The fourth saliency map has salient parts mostly near the flat parts (or what would be the flat parts) of the shape. For the third row or third club chair, the first saliency map has a relatively small number of salient parts. The second saliency map has salient parts near the edges or corners. The third saliency map has salient parts almost everywhere. The fourth saliency map has salient parts mostly near the flat parts of the shape.

The results for tables are shown in Figure 3. For the first table,

the first saliency map (for classifying the shape) has a relatively small number of red or salient parts. The second saliency map (for deciding shape aesthetic) has salient parts mostly around the edges (although less salient parts here as this shape is smaller overall) The third saliency map (for describing shape) has salient parts almost everywhere (especially if taking into account the rotational symmetry). The fourth saliency map (for deciding the materials the shape can be made of) has salient parts mainly near the flat parts of the table. For the second table, the first saliency map has a relatively small number of salient parts. The second saliency map has salient parts near the edges (although this shape has many edges and curves). The third saliency map has salient parts almost everywhere (especially if taking into account the rotational symmetry). The fourth saliency map has salient parts mostly near the flat parts. For the third table, the first saliency map has a relatively small number of salient parts. The second saliency map has salient parts mainly near the edges or corners. The third saliency map has salient parts almost everywhere. The fourth saliency map has salient parts mostly near the flat parts.

In Figure 4, for the first lamp, the first saliency map (for classifying the shape) has a relatively small number of red or salient parts. the second saliency map (for deciding shape aesthetic) has salient parts mostly near the edges or curved regions. the third saliency map (for describing shape) has salient parts almost everywhere. the fourth saliency map (for deciding the materials the shape can be made of) has salient parts mostly near the flat regions. For the second lamp, the first saliency map has a relatively small number of salient parts. The second saliency map has salient parts near the edges or curved regions. The third saliency map has salient parts almost everywhere. The fourth saliency map has salient parts mostly near the flat regions. For the third lamp, the first saliency map has a relatively small number of salient parts. The second saliency map has salient parts near the edges or curved regions. The third saliency map has salient parts almost everywhere. The fourth saliency map

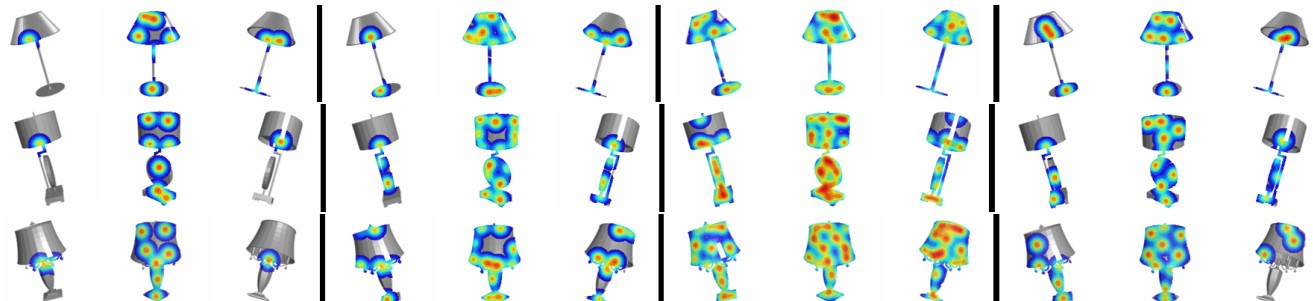

Figure 4: Results for 3 lamps: Each row shows the results for a lamp. Each column (separated by the vertical black lines) shows the 3 views of the 3D shape. The 4 columns show the saliency maps for the 4 tasks (see Figure 2 or the text for more details).

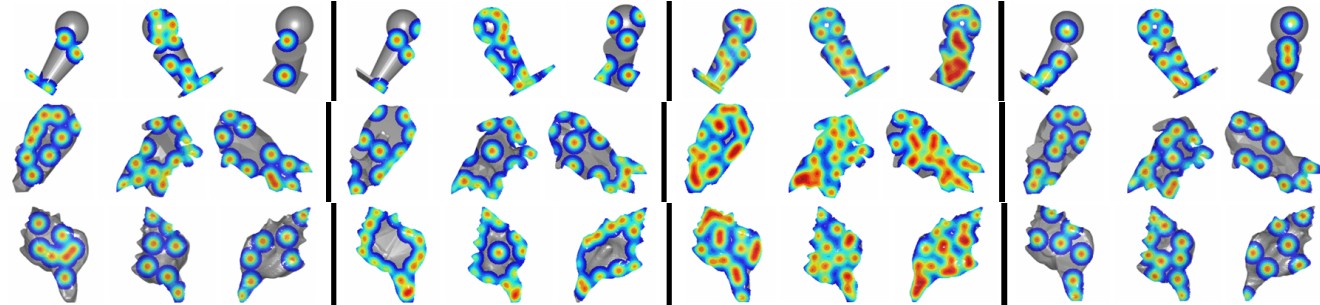

Figure 5: Results for 3 shapes from the "abstract" class: Each row shows the results for one shape. The 3 shapes are: a trophy, a statue of a dog, and a seashell. Each column (separated by the vertical black lines) shows the 3 views of the 3D shape. The 4 columns show the saliency maps for the 4 tasks (see Figure 2 or the text for more details).

has salient parts mostly near the curved regions (although the shape has mostly curved regions).

The results for "abstract" shapes are shown in Figure 5. For the first "abstract" shape of a trophy, the first saliency map (for classifying the shape) has a relatively small number of red or salient parts. The second saliency map (for deciding shape aesthetic) has salient parts mostly around the edges or curved regions. The third saliency map (for describing shape) has salient parts almost everywhere. The fourth saliency map (for deciding the materials the shape can be made of) has salient parts mostly near the curved regions (although the shape also has mostly curved regions). For the second "abstract" shape of a dog statue, the first saliency map has a relatively small number of salient parts (although more here as this shape is more complex). The second saliency map has salient parts near the edges. The third saliency map has salient parts almost everywhere. The fourth saliency map has salient parts mostly near the edges (although this shape has mostly edges). For the third "abstract" shape of a seashell, the first saliency map has a relatively small number of salient parts (although more here as this shape is more complex). The second saliency map has salient parts near the edges. The third saliency map has salient parts almost everywhere. The fourth saliency map has salient parts mostly near the edges (although this shape also has mostly edges as it is a complex shape).

We make some general observations from the above descriptions. In summary, for the task of classifying the shape, the task-aware saliency map has a relatively small number of salient parts (relative to our other three tasks). It seems that a relatively small number of parts need to be looked at in order to decide the shape class or type. For the task of deciding the shape aesthetic, the task-aware saliency map has salient parts near the edges or curved regions. This makes sense as curved parts are known to cause a shape to be more aesthetic [12, 28]. For the task of describing the shape, the

task-aware saliency map has salient parts almost everywhere. It seems the viewers decided to look almost everywhere to describe it as much as possible. For the task of deciding the materials the shape can be made of, the task-aware saliency map has salient parts on or near the flat regions (that could perhaps be made of a solid material), except when the shape is mostly curved (in which case the salient parts are located sparsely on the curved regions).

The main overall result from the above observations is: we have demonstrated that where people look on the surfaces of 3D shapes as they perform different tasks is different.

## 5 DISCUSSION, LIMITATIONS, AND FUTURE WORK

As described in the title of this paper, our goal was to demonstrate that the "task-aware saliency" concept is also true for 3D shapes. The "task-aware saliency" concept is that where people look on the surfaces of 3D shapes is different if they are performing different tasks as they look at the shapes. In this paper, we have successfully achieved our goal.

We believe that this goal is already an important contribution as a paper. We do not have any learning method (as too many papers do nowadays) to compute this task-aware saliency. The reason is that we wish to verify that this task-aware saliency concept is also true for 3D shapes first, as an important first step for this overall topic of task-aware saliency for 3D shapes.

Since we explored the topic of task-aware saliency for 3D shapes, we have limitations and can do future work in each of the categories of "task", "3D shapes", and "saliency".

For "task", our limitation is that we currently have four tasks. For future work, we can have more tasks, and we can confirm our results with more tasks.

For "3D shapes", our limitation is that we currently have four shape classes of three shapes each. For future work, we can have

more shape classes and more shapes, and we can then strengthen our results. Furthermore, we consider only the 3D geometry now, and there are no colors or textures on the shapes. As the color or texture could also affect where people look on the 3D shapes, future work can include these aspects of the shapes.

For the 3D shape representation, we collect data with a multiple-view image representation for now. For future work, we can explore more shape representations. We can have an automatically rotating 3D shape to allow participants to observe the shape from all views. We can also have an interface to allow participants to rotate/zoom/pan a 3D shape interactively as they look at the shape.

For "saliency", we display the saliency maps as multiple-view images for now. For future work, we can project the saliency values from the images to the 3D shape, and display saliency maps that are more like the traditional mesh saliency maps [21, 22].

Moreover, we could have more methods to collect saliency data. Eye-tracking devices are common for recording where people look, so we could use them for future work. However, our clicking interface [18] has been shown to correspond to eye-tracked locations, so we might expect the eventual results to be the same.

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
