# OpenReview forum: "Demonstrating Task-Aware Saliency for 3D Shapes"
_graphicsinterface.org/Graphics_Interface/2022/Conference — Submitted to GI 2022_

### Official Review · Reviewer_3k7n · 2022-04-13
**Useful and important study; setup not too valid, references missing**

**Rating:** 4
**Confidence:** 3

**Review:**

The paper presents a study into how humans look at renders of 3D object given different tasks. Guided by a very plausible hypothesis that the given task influences where humans look, they present subjects with renders of 3D objects, give them tasks (classify, describe how aesthetic the shape is, etc.), and implement a mechanism for tracking visual attention. They then present 'task-aware saliency maps' and discuss them qualitatively.

While the conclusions of the paper are purely qualitative and not necessarily novel, that part I think I can agree with. I also agree that this is an important subject, that it is highly likely that humans do rely on tasks to guide their visual attention in looking at 3D objects -- a hypothesis that has been verified time and again in other setups. Overall, the paper is written well and can be useful.

However, I see two drawbacks of the current submission that do not allow me to give a higher rating:
1. A whole class of relevant literature is left out. In HCI, this hypothesis has been studied in numerous contexts; the current paper fails to reference those. In fact, I think those might be more related than actual saliency papers, even though the maps look visually similar. For an example, please see

Brandon Victor Syiem, Ryan M. Kelly, Jorge Goncalves, Eduardo Velloso, and Tilman Dingler. Impact of Task on Attentional Tunneling in Handheld Augmented Reality. In Proceedings of the 2021 CHI.

and references therein.

2. Most importantly, however, I do not understand the reasoning behind the somewhat cumbersome visual attention setup. This blurring-clicking-deblurring procedure seems to have very poor validity -- it is very different from the act of looking. The first problem is that people don't have to click on everything they look at: For instance, humans might study a heavily blurred area first, e.g. a contour that is blurred but somewhat distinguishable, and decide they don't need to click there.
Furthermore, when people look, while the sharp circle may simulate foveal vision somewhat, we see outside fovea as well. I would expect the eye tracker to be less clustered, i.e. some click locations will be covered only by eye saccades.
I am not sure why the authors preferred this setup that lowers immensely the internal validity of the study to a tradition eye-tracking setup.

Overall, I'm currently not in favor of acceptance due to those issues.

---

### Official Review · Reviewer_qU5v · 2022-04-14
**methodology clearly below GI standard**

**Rating:** 1
**Confidence:** 4

**Review:**


This paper presents what appears to be a perceptual study mean to prove the hypothesis that “the task will influence where people look in 3D”.

I confess I do not understand the motivation behind this hypothesis and no background is given.

Moreover the methodology is filled with problems: appears that there are very few examples, the tasks and the procedure for the user is not described in any kind of detail. (who are the users, what are the tasks, etc.)

All in all this paper is clearly below the GI standard.

---

### Official Review · Reviewer_XhKV · 2022-04-14
**This work seeks to show that people look at different parts of the shape when solving for different tasks.**

**Rating:** 4
**Confidence:** 3

**Review:**

This work sets out a clear objective: 'Do people look at different parts of 3D objects when performing different tasks?'. The experiments with only 15 participants does show a tendency of users clicking at different regions and at different frequency depending on the task. However, it is not clear to me how this observation is any different in the 3D domain from the established observations in 2D. While the paper is relatively clear to read and follow, Section 4.2 present the findings in a mundane and long description that could benefit from brevity and a more concise exposition relating things to the visual evidence in the figure. I would recommend the paper for rejection at its current state and would note several points of feedback:
- A larger sample size might reveal more clear observations per class on what exactly users are looking at when solving for each task. The current results look widely distributed and its unclear how many people clicked on what region.
- A stronger motivation on why this problem is important to study. As it stand, I did not see a strong motivation behind the main question studied in this work.
- More details on the methodology of the study. Where users asked to solve the different tasks one after the other for the same shape or at random?
- This work seems to me more on the HCI or psychological side than on the graphics side. I have to declare that I find the topic to be far from my expertise so I could be mistaken in my judgment.

---

### Decision · Program_Chairs · 2022-04-17

Reject